# A Review of the Phytochemistry and Pharmacological Properties of the Genus Arrabidaea

**DOI:** 10.3390/ph15060658

**Published:** 2022-05-25

**Authors:** Jessyane Rodrigues do Nascimento, Amanda de Jesus Alves Miranda, Felipe Costa Vieira, Carla Daniele Pinheiro Rodrigues, Luna Nascimento Vasconcelos, José Lima Pereira Filho, Auxiliadora Cristina Corrêa Barata Lopes, Marcelo Marucci Pereira Tangerina, Wagner Vilegas, Cláudia Quintino da Rocha

**Affiliations:** 1Institut of Biosciences, Coastal Campus of São Vicente, Universidade Estadual Paulista-UNESP, São Vicente 11330-900, SP, Brazil; jessyane.nascimento@unesp.br (J.R.d.N.); vilegasw@gmail.com (W.V.); 2Department of Chemistry, Federal University of Maranhão-UFMA, São Luís 65080-805, MA, Brazil; amanda.quimica14@gmail.com (A.d.J.A.M.); felipe.cv@discente.ufma.br (F.C.V.); krla.pinheiro12@gmail.com (C.D.P.R.); luna.n.vasconcelos@gmail.com (L.N.V.); jlp.filho@outlook.com (J.L.P.F.); 3Biodiversity and Biotechnology Graduate Program-BIONORTE, Federal University of Maranhão-UFMA, São Luís 65080-805, MA, Brazil; auxiliadorabarata@hotmail.com; 4Botany Department, Institute of Biosciences, University of São Paulo-USP, São Paulo 05508-090, SP, Brazil; marcelomptang@gmail.com

**Keywords:** chemical composition, pharmacological activity, Arrabidaea, Cerrado

## Abstract

The genus Arrabidaea, consisting of ~170 species, belongs to the family Bignoniaceae, distributed around the Neotropics and temperate zone. The center of diversity of the family is in Brazil, where 56 genera and about 340 species exist. Most species of the genus Arrabidaea are traditionally utilized as diuretics and antiseptics, as well as for treating intestinal colic, diarrhea, kidney stones, rheumatoid arthritis, wounds, and enterocolitis. The genus is chemically diverse with different substance classes; most of them are triterpenes, phenolic acids, and flavonoids, and they exhibit valuable pharmacological properties, such as antitumor, antioxidant, leishmanicidal, trypanocidal, anti-inflammatory, and healing properties. This review presents information on the chemical constituents isolated from seven Arrabidaea species, and the pharmacological activities of the extracts, fractions and pure substances isolated since 1994, obtained from electronic databases. The various constituents present in the different species of this genus demonstrate a wide pharmacological potential for the development of new therapeutic agents, however its potential has been underestimated.

## 1. Introduction

Bignoniaceae species are widely utilized for ornamental purposes owing to the beauty of their flowers (e.g., Ipê), as well as for civil construction and the production of musical instruments owing to the hardness of the wood [1]. 

Brazil represents the diversity center of the family, where 56 genera and ~340 species, including many endemic taxa, occur. The Bignoniaceae species are found in different kinds of environments ranging from the Cerrado to the rain and evergreen forests, thus representing the main family of lianas in Brazilian forests. The species from this taxon, whose jacarandás (*Jacaranda brasiliana*), as well as yellow and purple ipês (*Tabebuia alba* and *Tabebuia avellanedae*), are the most representative examples of the family, are distributed in tropical regions globally, although they are more common in the American continent [2].

The genus Arrabidaea has a complex taxonomy since many characteristics are shared with species of other genera. The species can be a sub-shrub, a shrub, and even an upright tree with floral branches of 35–50 cm (the youngest being slightly flattened), with leaves with petioles of 5 to 10 mm, and sometimes 12 mm, as well as a simple blade. Further, it exhibits terminal paniculate inflorescence in the axillary branches, and flowers with small tubular chalice topped by five teeth and a bell-shaped corolla. The flowers exhibit pink to purple corollas, are rarely white, and are never yellow [3]. The genus Arrabidaea for over a century was the largest genus within the family Bignoniaceae, however the number of species varied due to new classification systems as a result of variation in the calyx, coralla, fruit, seed and inflorescence morphology [4,5,6].

There are little data available in the literature on the chemotaxonomy, chemical composition and biological activities; *Arrabidaea chica* (Bonpl.) Verl., is the most investigated species, reporting the presence of the chemical constituents phytosterols, flavonoids [7], and pigments such as carajurone, carajurine [8], and 3-desoxyanthocyanidins [9], utilized in cosmetics. Furthermore, some studies focused on species, such as *Arrabidaea samydoides* (Cham.) Sandwith, *Arrabidaea pulchra* (Cham.) Sandwith, *Arrabidaea triplinervia* (Mart. ex DC.) Baill., *Arrabidaea bilabiata* (Sprague) Sandwith, *Arrabidaea patellifera* (Schltdl.) Sandwith, and *Arrabidaea brachypoda* Bureau, because of their well-known utilization as antiseptics and diuretics, as well as for the treatment of wounds, intestinal colic, diarrhea, and enterocolitis. The phytochemical and pharmacological studies of these species have also revealed the antifungal [7], antimalarial [10], antitumoral [1], gastroprotective [11], and anti-*Trypanossoma cruzi* [12] activities of the extracts and isolated compounds, thus elucidating the bio-pharmacological potentials of this plant genus. 

In Brazilian ecosystems, particularly in the Cerrado, the genus Arrabidaea occurs in great numbers, and in the southeast region, some species are widely used in traditional medicine, but little is known by the scientific community of their natural chemistry products. Only seven of them have been investigated from a chemical and pharmacological point of view.

The Cerrado is the second largest biome in South America; it is greatly significant as a source of alternative medicines for treating different diseases. Although the flora of the Cerrado is extremely rich, only 1.5% of its extension is protected by law [9]. The deforestation rate in this biome exhibited an increase of 13%, totaling a suppessed native vegetation area of 7340 km^2^, in the year 2020 alone [13].

In this sense, the present review critically evaluates the biological, pharmacological, and chemical potential data gathered from the literature for seven species of the Arrabidaea genus, in only species chemically and biologically examined, in order to investigate the species from this biome that have been facing a fragmentation process and the risk of becoming extinct without knowing their chemical-pharmacological properties. 

## 2. Materials and Methods

A systematic literature review was conducted by searching three electronic scientific journal databases: Scielo, Science Direct, and PubMed. A search filter was applied to delimit the period between the years 1994 and 2021, and three sets of keywords were utilized: “*Arrabidaea* and *Fridericia*”, “*Arrabidaea* or *Fridericia*”, and “*Arrabidaea* and pharmacological applications”. The primary search identified 747 results, of which the articles were selected by excluding conference papers, course completion papers, scientific reports, drafts, and articles that were indexed in two or more databases. The selected articles were manually reviewed to identify and exclude articles that did not meet these criteria, and in the end, 56 articles were selected for this review on the genus *Arrabidaea* syn. *Fridericia*. Figure 1 illustrates the selection process. 

## 3. Results and Discussion

### 3.1. Main Bio-Pharmacological Properties and Chemical Composition of the Species from the Genus Arrabidaea

#### 3.1.1. *A. samydoides*

Synonyms: *Arrabidaea samydoides* (Cham.) Sandwith, *Fridericia samydoides* (Cham.) L.G. Lohmann, *Arrabidaea anguillulicarpa, Bignonia samydoides* (Cham.), *Bignonia varians* (DC.), *Panterpa varians* (DC.) Miers.

##### Main Chemical Composition of *A. samydoides*

The chemical composition from the leaves, and stems of *A. samydoides* included lupeol (**1**), sitosterol (**2**), stigmasterol (**3**), chrysin (**4**), 3β,16α-dihydroxy-olean-12-en (**5a**), erythrodiol (**5b**), uvaol (**6a**), and ursolic acid (**6b**), 2-(2′-*O*-trans-caffeoyl)-*C*-β-D-glucopyranosyl-1,3,6,7-tetrahydroxyxanthone (**7a**), 2-(2′-*O*-trans-cinnamoyl)-*C*-β-D-glucopyranosyl-1,3,6,7-tetrahydroxyxanthone (**7b**), and 2-(2′-*O*-trans-coumaroyl)-*C*-β-D-glucopyranosyl-1,3,6,7-tetrahydroxyxanthone (**7c**), mangiferin (**7d**), 2-(2′-*O*-benzoyl)-*C*-β-D-glucopyranosyl-1,3,6,7-tetrahydroxyxanthone (**7e**), muraxanthone (**7f**) [1,10] (Table 1).

##### Bio-Pharmacological Properties of *A. samydoides*

A representative species of the medicinal potential of this genus is *A. samydoides.* In 2003, researchers reported the chemical and pharmacological studies of this species. The subsequent fractionation and purification of the bioactive components demonstrated a gradual decrease in the bioactivity of the sub-fractions compared with the inhibition values that were determined by the crude extract in the evaluation of the activity toward repairing of DNA [1]. The authors attributed this activity loss to factors which were related to the synergistic effect resulting from the chromatographic separation of substances (a common result in bio-guided chemical studies). The compounds isolated from the stem and leaves of *A. samydoides* included (**1**), (**2**), (**3**), (**4**), (**5a**), (**5b**), (**6a**), and (**6b**) [1]. By focusing on the detection of the antitumoral activity during the screening tests in the DNA repair, the raw extract of the stem of *A. samydoides* exhibited inhibitory activity against the growth of mutant strains of Saccharomyces cerevisiae in the rad+ and rad 52Y lines with values of 12.0 and 13.0 mm, respectively, and was inactive toward the RS 321 lineage. Further, the CHCl_3_ fraction of the extract exhibited inhibitory activity against the rad+ (11.0 mm), and rad 52Y (12.0 mm) lines, while the purified substances of the bioactive fractions demonstrated a gradual decrease in their bioactivity [1]. 

In another study, Bolzani and collaborators reported the isolation of three C-glucosylxanthones, (**7a**), (**7b**), and (**7c**), from the stems of *A. samydoides.* Furthermore, they also isolated three known glucosylxanthones, (**7d**), (**7e**), and (**7f**). The xanthones, (**7a**) and (**7f**), exhibited moderate activity against free radicals through the reduction of 1,1-diphenyl-2-picrylhydrazyl (DPPH), as well as an antioxidant activity, which was confirmed from its properties that were measured via high-performance liquid chromatography equipped with an electrochemical detector (HPLC–ElCD) [10]. 

The antioxidant activity of *A. samydoides* was also reported in Part 2 of the study by [17]. They reported the antiviral activities of the three strains (VHH-1, EMCV, and VACV) that were employed in the study with EC_50_ values approximating that of the drug aciclovir. In the same study, the antioxidants, *C*-glucosyl xanthone, acylated derivatives, and mangiferin, were isolated from the species (*A. samydoides*), and the latter was observed via HPLC-diode-array detection (DAD) as the major constituent of the leaf samples of *A. samydoides*. Mangiferin is known for its potent antiviral activity, although the results observed in *Arrabidaea* might be related to the presence of another compound in the plant since isolated mangiferin is less active (EC_50_ = 218.1 + 3.4 μg/mL) [17].

#### 3.1.2. *A. pulchra*

Synonyms: *Arrabidaea pulchra* (Cham.) Sandwith, *Fridericia pulchella* (Cham.) L.G. Lohmann, *Arrabidaea tobatiensis*, *Bignonia pulchella* (Cham.), *Cuspidaria pulchella* (Cham.) Baill, *Cuspidaria pulchella* (Cham.) K. Schum. *Lundia pauciflora*, *Lundia pauciflora* Mart. ex DC., *Mansoa schwackei* Bureau & K. Schum., *Paracarpaea pulchella.*

##### Main Chemical Composition of *A. pulchra*

The most abundant constituents in *A. pulchra* are phenylethanoid glycosides verbascoside (**8a**), acteoside (**8b**), caffeoylcalleryanin (**9**), 6-hydroxyluteolin-7-*O*-β-glucoside (**10**) 3,4-dihydroxybenzaldehyde (**11**), *p*-coumaric acid (**12**), *p*-hydroxybenzoic acid (**13**), oleanolic acid (**14**) (Table 2) and ursolic acid (**6a**), also present in *A. samydoides* [14,15,16,18].

##### Bio-Pharmacological Properties of *A. pulchra*

The *Arrabidaea pulchra* (Cham.) Sandwith belongs to the *Bignoniaceae* family, which is a liana exhibiting membranaceous, chalice with multicellular trichomes and a reddish glandular apex distributed throughout the inflorescence, thus imparting this species with a brownish color [3]. In Brazil, plants belonging to this family are distributed from the Amazon to the Rio Grande do Sul; they do not exhibit a unique habitat.

A phytochemical study of the ethanolic extract of the leaves of *A. pulchra* revealed that tannins and flavonoids were its major constituents; the literature has reported the significant antitumoral activities of these substances. The bioassay of the cell viability of the extract exerted an in vitro cytotoxic effect on the concentrations of 500 and 100 μg/mL after 24 h of incubation under the cells of the PC-3 lineage (prostate cancer). It also exhibited an antiproliferative activity of >50% of the cytotoxic effect of the doxorubicin concentration (5 μg/mL) that was employed as a positive control, thus representing a cytotoxic effect, which was superior to the standard drug [19].

An antiviral screening study of the species belonging to the *Bignoniaceae* family occurring in Minas Gerais revealed that the ethanolic extract of *A. pulchra* leaves exhibited relevant antiviral activity compared with those of 34 extracts of 18 plant species that were selected based on ethnopharmacological and taxonomic criteria [17]. 

The activity of the extract of *A. pulchra* was evaluated against human herpesvirus (HSV-1) and Western Reserve virus Vaccinia (VACV-WR), as well as RNA viruses, murine encephalomyelitis virus (EMCV), and dengue virus (DENV-2), by the colorimetric assay of 3-(4,5-dimethylthiazol-2-yl)-2,5-diphenyltetrazolium (MTT). Its cytotoxicity was determined in LLCMK2, after which the Vero cells and selectivity indexes (SI) were calculated. The extract exhibited activity against DENV-2 (EC_50_ = 46.8 ± 1.6 μg/mL) and SI of 2.7 [14]. 

Compounds (**8a**) and (**6a**) exhibited similar anti-DENV-2 profiles, with SI values of 3.8 and 3.1, respectively. Moreover, (**8a**) and (**9**) inhibited the replications of DENV-2 with EC_50_ values of 3.4 and 2.8 μg/mL, respectively, although (**9**) exerted a less cytotoxic effect on the LLC-Mk2 cells, resulting in a more favorable SI (20.0). However, regarding (8a), the SI value (3.8) was approximately five times lower. Compound (**9**) was the most effective anti-DENV-2 constituent, exhibiting 100% inhibition at 10 μg/mL [14]. 

The authors evaluated the lipoxygenase (LOX) inhibitory activities of the n-hexane, ethyl acetate (EtOAc), n-butanol, and hydromethanolic fractions of *A. pulchra* ethanolic extract. The n-butanol fraction obtained the best inhibition value of the enzyme, 15-LOX, from 62.8 to 25 µg/mL, representing the most relevant inhibition within the assay. Compounds 9 and 8b exhibited significant inhibitory activities with IC_50_ values of 1.59 and 1.76 uM, while the utilized positive control (Zileuton) obtained a value of 1.54 uM [18]. The cytotoxicity assay was performed for (**9**), (**8b**), and (**10**) in the cell lines, GM07492-A (human lung fibroblasts), B16F10 (murine melanoma), and HepG2 (human hepatocellular carcinoma). Compound (**9**) did not exert a cytotoxic effect with a CC_50_ value of >800 uM, while (**8b**) reduced the cell viability of the B16F10 and HepG2 lines (CC_50_ = 176.8 and 271.2 µM, respectively). Compound (**10**) exhibited CC_50_ values of 513.0, 499.0, and 283.2 µM for GM07492-A, B16F10, and HepG2, respectively, indicating that (**8b**) and (**10**) exhibited cell selectivity [18]. The authors also evaluated the in vitro schistosomicidal activities of the hexane (FH), buthanolic, hydromethanolic, and EtOAc fractions of the areas of the *A. pulchra* extract against adult worms, *Schistosoma mansoni*. The in vitro test revealed that the incubation of *S. mansoni* worms in 100 μg/mL ethanolic extract of *A. pulchra* induced the deaths of 25% of the worms within 120 h and reduced the motor activities of the others; however, no significant integumentary changes were identified. EAA demonstrated the highest effectiveness among the tested fractions, whereas the incubation (100 μg/mL) for 120 h accounted for 50% of the deaths and a visible reduction in motor activity, while the FH, buthanolic, and hydromethanolic fractions exhibited inactivity. The following five substances were isolated from ethanol extract of leaves: (**11**), (**12**), (**13**), (**6a**), and (**14**). Compounds (**11**) and (**12**) could only separate coupled adult *S. mansoni* worms at 100 µM [15].

#### 3.1.3. *A. triplinervia*

Synonyms: *Arrabidaea triplinervia* (Mart. ex DC.) Baill., *Fridericia triplinervia* (Mart. ex DC.) L.G. Lohmann, *Arrabidaea ateramnantha* Bureau & K. Schum, *Arrabidaea triplinervia* var. brachycalix Hassl., *Bignonia triplinervia* Mart., *Bignonia triplinervia* Mart. ex DC., *Petastoma triplinervia* (Mart. ex DC.) Miers, *Saritaea triplinervia* (Mart. ex DC.) Dugand.

##### Main Chemical Composition of *A. triplinervia*

The chemical composition of the hydroethanol extracts of *A. triplinervia* leaves demonstrated the presence of three triterpenes, namely ursolic acid (**6a**), oleanolic acid (**14**), pomolic acid (**15**), and alpinetine (**16**) (Table 3) [16].

##### Bio-Pharmacological Properties of *A. triplinervia*

The ethanolic extract of *A. triplinervia* leaves exhibits significant trypanocidal activity in vitro against *Trypanosoma cruzi* trypomastigotes, the etiological agent of Chagas’ disease, causing the total elimination of the blood parasites at a concentration of 5.0 mg/mL; a decrease of ~50% of the parasites was observed at 2.5 mg/mL [16]. The ursolic and oleanolic acids exhibited significant trypanocidal activities that resulted in the total elimination of active parasites at concentrations of 0.4 and 1.6 mg/mL, respectively. Alpinetine was inactive at the tested concentrations, while pomolic acid was not tested owing to the availability of an insufficient quantity. The structure–activity relationships (SAR) of a series of synthetic compounds from the ursolic acid derivatives and oleanolic acid were studied, and the presence of the free hydroxyl and/or carboxylic groups is necessary for the trypanocidal activity since they could be deduced from the effects of acetates, methyl esters, and the aldehydes derivatives [16].

#### 3.1.4. *A. bilabiata*

Synonyms: *Arrabidaea bilabiata* (Sprague) Sandwith, *Tanaecium bilabiatum* (Sprague) L.G. Lohmann, *Adenocalymna bilabiatum* (Sprague) Sandwith, *Arrabidaea cuminaensis* A. Samp., *Arrabidaea kuhlmannii* A. Samp., *Memora bilabiata* Sprague, *Memora nobilis* Miers, *Pseudocalymma kuhlmannii* (A. Samp.) J.C. Gomes.

##### Main Chemical Composition of *A. bilabiata*

The chemical constituents isolated in *A. bilabiata* were monofluoroacetic acid (**17**), (**3**) and (**6a**), as well as allantoin (**18**) [20,21]. The secondary metabolites were found mainly in the leaves, seeds and aerial parts of the plant (Table 4).

##### Bio-Pharmacological Properties of *A. bilabiata*

*Arrabidaea bilabiata*, popularly known as “chibata” or “gibata”, is an essential toxic *Bignoniaceae* that accounts for many cattle deaths, which occur in the extensive floodplain regions of the Amazon basin [23]. *Arrabidaea bilabiata* and *Palicourea marcgravii* exhibit potentially lethal effects on the animals after ingestion, requiring 1–10 min between the appearance of the first symptoms and the occurrence of eventual death. The toxic principle, (**17**), was isolated from both species; *A. bilabiata* exhibited the highest concentration in seeds by a factor of 21 [21].

In the authors’ experiments, the sensitivities of buffalo and cattle to *A. bilabiata* were verified, as well as the difference between toxicities of the new and ripe leaves at different periods of the year (end of dry and rainy seasons). It was verified that, although the clinical–pathological picture was essentially the same, the buffalo demonstrated more resistance to the toxic action of *A. bilabiata*, at least two times that of cattle. The experiments also demonstrated that the new leaves of this plant were two times as toxic as the ripe leaves, while October represented the greatest toxicity. Moreover, the lowest doses that caused the deaths of the animals were 3 and 2 g/kg new leaves for the buffalo and cattle, respectively. These data also indicated that the lower incidence of intoxication by the plants that caused sudden deaths in the buffalo in the Amazon was possibly due to the greater resistance of this animal species since their preferred habitat is the floodplain regions, the habitat of *A. bilabiata* (a plant, which is less toxic than *Palicourea marcgravii* St. Hil. that is found on terra firm, the preferred habitat for cattle) [24].

Proceeding with the studies on the toxicity of *A. bilabiata*, the authors [23] conducted a study employing mature leaves and sprouts, which were collected in the same place and at the same periods of flood and drought employing rabbits because it is a species that is sensitive to intoxication by this plant. In the experiments employing sprouts that were collected in October, the lowest dose, which caused the deaths of the rabbits, was 0.5 g/kg; in May, it was 1.0 g/kg. Conversely, the lowest doses for the mature leaves that were collected in October and May were 4.0 and 6.0 g/kg, respectively, further proving that the new leaves, which were collected in October, were more toxic than the ripe ones. This confirms the data that were previously obtained from cattle and buffaloes [23].

Monofluoroacetic acid (MF) is one of the most toxic substances already discovered and has been isolated from several plants; its ingestion causes “sudden death” of several animals, leading to significant economic losses. Studies have revealed that *A. bilabiata* is responsible for the deaths of several cattle. The administration of acetamide to animals that were intoxicated by MF exerted considerable protective effects. The protective effect of acetamide on rat poisoning by MF and the aqueous extracts from eight toxic plant species, including *A. bilabiata*, was evaluated. When previously administered in sufficiently high doses, acetamide could prevent the appearances of medical signs or the death of all MF-intoxicated mice, as well as those intoxicated by fresh leaves of *P. marcgravii* or with concentrated extracts of *Palicourea marcgravii, Palicourea juruana*, *Palicourea elegans*, *Arrabidaea bilabiata*, *Arrabidaea rigida*, and *Arrabidaea exotropica*, confirming the presence of MF and availing subsidies for the development of genetic studies in Brazil involving the metabolization of MF by rumen bacteria [25]. 

However, some authors have argued that MF would not be the only toxic active principle that determined the deaths of animals that ingested the plant; they opined that there might be other compounds that could cause the deaths of animals and exert a synergistic effect with MF to contribute to the toxicity of the species. To isolate the active principle, the dichloromethanolic, methanolic, and isopropanolic extracts of the aerial parts of *A. bilabiata* were prepared. These extracts were administered orally to rats (doses = 0.5–2 mg/kg) to determine the substances that affected spontaneous motor activities. The isopropanolic extract significantly increased the long and short pause movements with values of 112% and 54%, respectively, while the methanolic extract increased motor activity but did not obtain statistical significance [20]. 

The chemical analysis ensured the isolation and identification of (**3**) and (**6a**) from the dichloromethane (DCM) extract, as well as (**18**) from the isopropanolic extract, because only (**18**) exhibited sufficient mass available to realign the motor activity test. At doses of 1, 2, 12, and 120 mg/kg, there was an increase in the prolonged pause movements but not for a short duration in the tested lower doses. However, in the highest dose tested (60 mg/kg), (**18**) could slightly reduce the motor activity in rats. Additionally, the in vitro activities of the extracts were analyzed against *Escherichia coli*, *Klebsiella pneumoniae*, *Pseudomonas aeruginosa*, *Staphylococcus aureus*, *Bacillus* sp., and *Candida albicans*. All the microbial strains demonstrated resistance toward the extracts except *C. albicans*, indicating its possible antifungal activity [20]. 

#### 3.1.5. *A. brachypoda*

Synonyms: *Arrabidaea brachypoda* Bureau, *Arrabidaea brachypoda* var. *platyphylla* (DC.) Bureau, *Fridericia platyphylla* (Cham.) L.G. Lohmann, *Alsocydia firmula* Mart., *Alsocydia firmula* Mart. ex DC., *Arrabidaea bangii* Sprague, *Arrabidaea elliptica* Bureau & Schumann, *Arrabidaea macrophylla* Schumann, *Arrabidaea platyphylla* (Cham.) Bureau & Schumann, *Bignonia brachypoda* var. *platyphylla* (Cham.) DC., *Bignonia platyphylla* Cham., *Bignonia regnelliana* Sond., *Petastoma simplicifolium* Miers.

##### Main Chemical Composition of *A. brachypoda*

Based on spectroscopic methods, a total of nineteen substances were isolated and characterized from leaves, flowers, aerial parts and roots of ethanolic hydroethanolic extract of *A. brachypoda*; namely, the flavonoids 3,4-dihydroxy-5,6,7-trimethoxyflavone (**19a**), cirsiliol (**19b**), cirsimaritin (**19c**), and hispidulin (**19d**) (Table 5), were isolated, indicating the first time that (**19b**) would be discussed in the *Bignoniaceae* family [7]. In addition, an oleanane-type triterpenoid, 3β-estearioxi-olean-12-eno (**20**), was isolated from the hydroethanolic extract of the roots [26]. The phytochemical investigation of the hydroethanolic root extract (HE) made possible the isolation of two glycosylated phenylethanoid derivatives (**21a** and **21b**), three rare novel dimeric flavonoids, namely brachydins 1 (**22a**), 2 (**22b**), 3 (**22c**), and unusual glycosylated dimeric flavonoids (**22d**–**j**), which were reported for the first time in the *Bignoniaceae* family [11,27,28,29]. Compounds (**23a**–**23p**) were derived from halogenation (bromination, iodination, and chlorination) reactions [12]. A phenylethanoid glycoside, namely conadroside (**24**) was isolated from leaves of *A. brachypoda* by Bertanha et al. [30]. The chemical characterization of the leaves extract revealed the presence of flavonoids apigenin (**25**), luteolin (**26**), isoquercitrin (**27**), rutin (**28**); furthermore, four chalcones were isolated from the flowers, namely 4′-hydroxy-3-4-dimethoxy-chalcone (**29**), 3′-hydroxy-3-acetate, 4- methoxy-chalcones (**30**), 3′,4′-dihydroxy, 3,4,4′-trimethoxy-chalcone (**31**), and 3,4-dimethoxy-chalcone (**32**) [31,32,33].

##### Bio-Pharmacological Properties of *A. brachypoda*

This is one of the species most studied by our research group. We have investigated extracts from the roots, leaves, and flowers. In 2014, we isolated for the first time a class of unusual dimeric flavonoids. This class of flavonoids was named brachydins.

*A. brachypoda* is a native plant from the Brazilian Cerrado; it is popularly known as “cipó-una” or “tintureiro”, and exhibits scandent bushes of up to 70 cm, possesses cylindrical branches, and is striated glabrous with lenticules. It is marked by the absence of pseudostipules. Its leaves are 1-2-foliolated with leaflets of 8–10 × 4–5 cm. The leaves are oblong and obovate, leathery, concolor, exhibiting acuminate and rounded acuminate apexes, sometimes mucronulate. It exhibits an entire border, that is flat with a cuneate base, palmated nervation, glabrous adaxial, and abaxial faces; its petioles are 1.5–2 cm, exhibiting simple tendrils. Its inflorescences are thyrsus with a glabrous inflorescence axis. Its epicalyx is absent, and it exhibits toothed calyx, is glabrous, and possesses pink to purple corolla with entirely white tracks. The infundibuliform is densely tomentosus, and the stamens include glabrous antennas and reduced staminodes that are glabrous; it exhibits a conspicuous nectary disk. The septifragal capsule is 21–25 × 1.5–2, which is coriaceus, glabrous, exhibits a persistent calyx, as well as wing membraneous seeds [3].

The antifungal activity of the raw CHCl_3_ extract, as well as the compounds that were isolated via the direct bioautography method on TLC plates, have been reported. The epicuticular waxes of the leaves of *A. brachypoda* were analyzed, and flavonoids, such as (**19a**), (**19b**), (**19c**), and (**19d**), as well as the first report of (**19a**) as a natural product (it had been reported as a synthesized compound that exhibited inhibitory activity against arachidonate 5-LOX) [37]. The bioassay comprising the suspension of *C. sphaerospermum* spores revealed that flavonoids (**19a**), (**19b**), and (**19c**) were potent fungi-toxic agents that inhibited fungus growth in 1 µg of the pure compound, while (**19d**) was the least active, exhibiting a growth inhibition value of 10 µg. It is believed that its epicuticular wax can act as a physical or chemical barrier against the attacks of microorganisms; the inhibition of the growth of the fungus, *C. sphaerospermum*, by the flavonoids that were isolated from the wax, *A. brachypoda*, is a crucial indication, which reinforces the ecological role of these compounds [7].

A study carried out with the ethanolic extract of *A. brachypoda* roots has been shown to exert anti-inflammatory and analgesic effects, thus supporting the traditional utilization of this plant in the treatment of some painful and inflammatory conditions. The chemical profile of an *A. brachypoda* root extract revealed that flavonoids and triterpenes are its main constituents. The oleanane-type triterpenoid (**20**) inhibited the two phases of the painful stimulation of the formalin test similarly to morphine. Further, it significantly inhibited the carrageenan-induced edema of rat paws at doses of 5, 10, and 15 mg/mL (p.o.), with 3 h post-carrageenan edema inhibitor values of 47.9%, 53.1%, and 50.4%, respectively [26]. 

The hydroethanolic extract of the *A. brachypoda* root (HEAB) exerted significant gastroprotective effects in vivo. The activity was evaluated via several experimental gastric ulcer models in rats (absolute ethanol, glutathione depletion, nitric oxide depletion, non-steroidal anti-inflammatory, pyloric ligation, and acetic acid). The extract significantly reduced the gastric lesions in all the models at a dosage of 300 mg/kg (p.o) and achieved the cytoprotection of the gastric mucosa [11].

A study evaluated the antileishmania activities of three rare novel dimeric flavonoids, namely brachydins (**22a**), (**22b**), and (**22c**), substances uncommon in the *Bignoniaceae* family-purified *A. brachypoda*. Brachydin (**22b**) was the most active against the promastigote form of *Leishmania* sp.; the in vitro quantification of the infected macrophages of *Leishmania amazonensis* revealed that (**22b**) was the most active against intracellular amastigotes but without exhibiting host cell toxicity. The combination of amphotericin B (the reference drug employed for conventional treatment) and the substance (**22b**) exerted an inductive effect, indicating the absence of any interaction between them. When administered to *L. amazonensis*-infected mice, both oral and topical treatments were ineffective against the parasite. Thus, brachydins might benefit from new studies for developing a new active structure against Leishmania [27].

The same authors achieved the halogenation of brachydins to increase their thermal and oxidative stabilities and evaluate their antiparasitic activity against *Trypanosoma cruzi* and *Leishmania amazonensis*. Thus, 16 compounds (**23a**–**23p**) were derived from the halogenation (bromination, iodination, and chlorination) reactions; among them, (**23g**) and (**23o**), which were derived from bromine and chlorine, respectively, demonstrated effectiveness against the amastigote form of *L. amazonensis*, besides exhibiting higher SI. They were more active than the original compounds. Regarding the activity, the *T. cruzi* compounds (**23e**), (**23k**), (**23n**), and (**23o**) exhibited the best IC_50_ values. Moreover, (**23a**) and (**23h**), which were obtained from the inactive derivative, demonstrated the highest activity, indicating that the bromine and iodine insertions could increase the biological activity of the original natural products [12].

LOX is an essential enzyme that is involved in inflammatory processes. Similar to *A. brachypoda*, its anti-inflammatory activity has been reported. The authors developed a procedure for directly identifying bioactive compounds in raw plant extracts, as well as evaluating the inhibitory activity of 15-LOX in vitro. Compound (**24**) achieved the inhibition of LOX with an inhibitory concentration of 7.8 µM, which is close to the standard, quercetin, with an IC_50_ value of 7.6 µM. Furthermore, conandroside was not cytotoxic to normal cells; it was confirmed as an LOX inhibitor; an evaluation of the LOX–conandroside complex exhibited a slightly higher docking score than quercetin [30]. 

The cytotoxicities of the dimeric flavonoids, (**22a**), (**22b**), and (**22c**), which were isolated from the *A. brachypoda* roots, were evaluated in human prostate tumor cells (PC-3) where they exhibited IC50 values of 23.41, 4.28, and 4.44 µM, respectively. Additionally, (**22b**) and (**22c**) increased the p21 levels that are related to the G1 cell cycle arrest; (**22a**) and (**22b**) decreased the expressions of phospho-AKT, which is responsible for cell cycle and proliferation. The two brachydins increased the levels of a protein that is related to the repair of DNA, as well as the induction of apoptotic processes [28].

The antispasmodic activity of the HE of the leaves (EH-FAB) of *A. brachypoda* was evaluated employing intestinal smooth muscle tissues (jejunum) through the carbachol-induced amplitude curves (CCH). EH-FAB inhibited the contraction and relaxation of isolated rat jejunum in a significant and concentration-dependent manner. The authors also observed the significant relaxation of the rat jejunum when it was precontracted by high concentrations of extracellular potassium, demonstrating the participation of the potassium channel in the relaxation mechanism of EH-FAB, and this further shifted the CaCl_2_ curve to the right, with a decrease in the maximal effect that is characteristic of the CaV inhibition of the calcium influx [31].

The chalcones (**29**, **30**, **31** and **32**) did not exhibit any antimicrobial activity on the tested *S. aureus* strains; however, all the chalcones increased the activity of NorA against the SA1199-B strain, and chalcone (**32**) exhibited the best modulatory effect and good ability to potentiate the action of EtBr against the SA1199-B strain. The molecular docking of the chalcones revealed that they could bind in the hydrophobic cavities of NorA and MepA, indicating that chalcone (**32**) competes with the antibiotic for the same NorA and Mep binding sites [32].

Compound (**26**) is a flavonoid with promising biological activities, including antitumor activity. This substance (**26**) was isolated from *A. brachypoda* leaves and tested on a panel of tumor cell lines in different tissues where it exhibited promising activity against all the tested cell lines, particularly U-251 glioblastoma, against which it exhibited good selectivity for the tumor cells compared with the positive control. It also controlled cell migration and tumorigenesis by inhibiting the IGF1 R/PI3K/AKT/mTOR pathway, causing the deaths of the U-251 tumor cells by apoptosis via the depolarization of the mitochondrial membrane [33].

*A. brachypoda* is a plant which is traditionally utilized to treat joint pains. The authors evaluated the in vitro anti-inflammatory activities of dichloromethane fraction (DCMAB); hydroethanolic extract (HEAB); and (**22a**), (**22b**), and (**22c**) in arthritis synoviocytes. DCMAB achieved 30% inhibition in the release of the pro-inflammatory cytokine IL-6 at 25 µg/mL, and HEAB was inactive; conversely, (**22b**) and (**22c**) decreased the release of IL-6 with 80% and 94% inhibitions, respectively, while (**22a**) did not achieve significant inhibitions at 25 and 50 µM with percentage values of 10% and 24% inhibition, respectively. The activity of DCMAB could be related to the higher amounts of (**22b**) and (**22c**), as revealed from the quantifications of the fractions: 108 and 123 mg/g, respectively [29].

#### 3.1.6. *A. chica*

Synonyms: *Arrabidaea chica* (Bonpl.) Verl., *Fridericia chica* (Bonpl.) L.G. Lohmann, *Adenocalymma portoricensis* A. Stahl, *Arrabidaea acutifolia* DC., *Arrabidaea chica* (Bonpl.) Verl., *Arrabidaea chica* F. cuprea (Cham.) Sandwith, *Bignonia chica* (Bonpl.), *Bignonia cuprea* (Cham.), *Bignonia rubescens* S. Moore, *Bignonia rufescens* DC., *Bignonia thyrsoidea* DC., *Bignonia triphylla* Willd. ex DC., *Lundia chica* (Bonpl.) Seem., *Temnocydia carajura* Mart. ex DC., *Vasconcellia acutifolia* Mart. ex DC.

##### Main Chemical Composition of *A. chica*

The guided fractionation of the hydroalcoholic *A. chica* extract (ACE) facilitated the isolation of four 3-deoxyanthocyanidins, namely 6,7-dihydroxy-5,4-dimethoxy-flavylium (**33a**, carajurin), 6,7,4′–trihydroxy-5-methoxy-flavylium (**33b**, carajurone), 6,7,3′,4′–tetrahydroxy-5-methoxy-flavylium (**33c**), and 6,7,3′-trihydroxy-5,4’-dimethoxy-flavylium (**33d**) [8] (Table 6). In addition, a new flavone, 6,7,3′,4′-tetrahydroxy-5-methoxy flavone (**34**), which is called carajuflavone, three flavonoids, vicenin-2 (**35**), kaempferol (**36**), and 4′-hydroxy-3,7-dimethoxyflavone (**37**), were reported for the first time in the genus *Arrabidaea* [38,39]. The chemical profile of the leaves allowed the isolation and identification of scutellarin (**38a**) and a rich fraction of the compound (**33b**), 6,7-trihydroxy-5-methoxyflavone (**39**), β-carotene (**40a**), α-carotene (**40b**), 6,7,3′,4′-tetrahydroxy-5-methoxyflavilium-*O*-glucuronide(**41a**), scutellarein-*O*-glucuronide (**41b**), and 5-methyl-scutellarein-*O*-glucuronide (**41c**) [34,36,40,41,42]. The compounds (**26**), (**19a**) and (**25**) have also been found in *A. chica*.

##### Bio-Pharmacological Properties of *A. chica*

The *Bignoniaceae*, *Arrabidaea chica*, known as “carajuru”, “pouca panga”, “chica”, or “pariri”, is a woody vine that is distributed around tropical America. The leaves of this plant are employed as an anti-inflammatory agent and astringent, as well as medicine for stomach aches, bloody diarrhea, leucorrhea, anemia, and leukemia in the Amazon region. The in vivo pharmacological inhibition of NF-kB of the isolated substances was evaluated to determine whether they substantially modulate the inflammatory processes exhibited by the extract. As observed, anthocyanidin (**33a**) accounted for the inhibition of the transcription factor, NF-kB, which is a phenomenon that is responsible for anti-inflammatory activities [8].

The biological tests of the isolated compounds revealed that (**34**) exhibited low inhibitory activity toward the release of enzymes and the production of superoxides in rabbit neutrophils, as simulated, employing formyl-methionyl-leucil-phenylalanine (FMLP), while (**26**) achieved the high inhibition of the neutrophil response effects [39]. 

An evaluation of the trypanocidal activity of the ethanolic extract of the leaves revealed its activity against the trypomastigote forms of *T. cruzi* at a concentration of 4 mg/kg at which it induced 41% cell lysis; this fraction (ethanolic extract) also induced 71.0% and 54% cell lysis at a concentration of 2 mg/mL [38].

The healing ability of an *A. chica* leaf extract was studied via in vitro tests, such as fibroblast growth stimulation and collagen production. The extract increased collagen production similar to vitamin C and (**18**), which were utilized as control. The in vivo test revealed that the extract could reduce 13% of skin lesions within only two days of topical application; moreover, it demonstrated 96% wound healing after 10 days, while the control group, which was treated with (**18**), exhibited 87% healing. These findings indicated that the extract was more efficient in wound healing than the positive control. The ethanol-induced ulcer healing activity of *A. chica* was also evaluated in rats in which the extract was administered orally, and the results demonstrated that it reduced the lesion rates by 76% and 90%, while the control reduced them by 96% [22]. 

The aqueous (EA) and ethanolic (EE) extracts of *A. chica* leaves were evaluated for their in vivo antitumor properties. The phytochemical analysis of the extracts revealed the presence of different classes of secondary metabolites, such as anthocyanidins, flavonoids, tannins, and saponins. At 30 mg/kg, the EE and EA reduced the development of Ehrlich solid tumor in the rats after ten days of oral administration. The reduction of the tumor size in the EE group was accompanied by an increase in the number of neutrophils in the blood. Conversely, the reduced growth of the tumor in the AE group did not change the number of neutrophils in the blood. Further, no adverse side effects were observed owing to the treatment [43].

The effects of the EA of *A. chica* on the edema caused by *Bothrops atrox* and *Crotalus durissus ruruima* snake venom were studied in orally, intraperitoneally, and subcutaneously treated albino rats. The extract inhibited the edema when it was subcutaneously and intraperitoneally administered with values of 55.87% and 65.70% for the genera Bothrops and Crotalus, respectively; the inhibition values were 79.81% and 92.52%, respectively, while the oral route exhibited inactivity probably because it was more susceptible to chemical transformations caused by enzymatic processes in the intestinal tract, which alters its inhibitory function. Additionally, histopathological analysis was employed to demonstrate the inhibition of granulocyte infiltration and myocytolysis, which were induced by a snake toxin [44].

The authors [45] employed in vitro anticancer activity models to evaluate the influence of the variation in the relative proportion of anthocyanidins in the methanolic extracts of *A. chica* leaves that were collected over 13 months and prepared with and without enzymatic treatments on a panel of nine human tumor cell lines, MCF-7 (breast), NCI-ADR/RES (ovary with multiple resistance phenotypes), UACC-62 (melanoma), NCI-H460 (lung), PC-3 (prostate), HT29 (colon), OVCAR-03 (ovary), 786-0 (kidney), and K562 (leukemia), while the antioxidant activity was evaluated via the DPPH test. The raw extract, which was not treated with xylanases, stimulated the growth of fibroblasts of the 3T3 lineage in concentrations of 0.25, 2.5, and 25 μg/mL between January and June. After the enzymatic treatment, the extracts exhibited in vitro antiproliferative activity between January and August. Furthermore, all the extracts reduced the effective cellular concentration of TGI by 250 μg/mL and induced fibroblast growth. In the DPPH tests, the untreated raw extract exhibited moderate antioxidant activity, while the treated extract improved the antioxidant activity. These results indicated that the presence of anthocyanidins favors this property [45].

In vivo assays were employed to evaluate the effect of foliar ethyl extract in Wistar rats [46]. Therein, the oral administration of the extract in doses of 300, 500, and 600 mg/kg for seven days inhibited carbon tetrachloride-induced liver damage; these results demonstrated a reduction of the serum levels of the glutamic-pyruvic transaminase enzymes by 85.34%, 88.59%, and 93.72%, while the levels of the oxaloacetic glutamic transaminase were 56.86%, 65.27%, and 68.95%, respectively. Additionally, the serum bilirubin levels were 83.81%, 83.12%, and 84.14%, which were similar to a 35 mg/kg dose of standard silymarin drug [46].

The antioxidant capacity of EE of *A. chica* leaves and their partitions in hexane, DCM, EtOAc, and butanol were evaluated at concentrations of 5, 10, 25, 50, 125, and 250 μg/mL. The DCM fraction exhibited better antioxidant capacity owing to its high concentration of (**26**). The in vivo assay employing Wistar rats was performed to evaluate the diuretic activity of the extract. For the assay, 25, 50, 100, and 200 mg/kg doses were administered, and furosemide was employed as the positive control (50 mg/kg). After the treatment, the urine volumes were measured for 6 h with collections every 2 h. The results demonstrated that FH (dose = 100 mg/kg) increased the urinary volume by 79.26% compared with the negative control (water). The treatment with (**26**) increased the volume by 93.92% when the same dose was administered [35].

The pharmacognostic analysis of the *A. chica* leaves was performed to facilitate the standardization and quality control of the plant material for application in natural products. The test to determine loss due to desiccation revealed a variation coefficient of 9.18%, while the determination of the total ash content was 6.07%. Further, the phytochemical test of the 70% hydroalcoholic dye of *A. chica* indicated the presence of the following classes of metabolites: reducing sugars, alkaloids, anthocyanidins, anthraquinones, anthocyanins, steroids, triterpenoids, phenols, flavonols, flavanones, saponins, and condensed tannins [47].

In 2013, some researchers obtained anthocyanins from *A. chica* leaves via sequential extraction in a fixed bed employing supercritical carbon dioxide (scCO_2_). The percentage of the general yields of the nonpolar scCO_2_ extract and CO_2_/ethanol mixture (80:20) were 0.65% and 2.40%, respectively. However, the addition of 6.0% water to the mixture increased the percentage yield to 50% in the latter. The test of the total phenols and flavonoids revealed that the extraction with CO_2_/ethanol (80/20) without water yielded an extract with a high concentration of gallic-acid-derived phenolic compounds. The hydroalcoholic extract (70:30), which was obtained via conventional methods, exhibited a higher concentration of phenolic compounds. The comparative chromatograms of the sequential extraction revealed that the extract, which was obtained at the third stage of the process employing water, considerably reduced the presence of anthocyanins, while the extraction employing the pure scCO_2_ solvent exhibited selectivity toward carajurine. Finally, the conventional ethanolic extract exhibited better values of carajurine concentration, even with a low total yield [48].

The cytotoxic potential of 70% HE of the leaves of *A. chica* was investigated in the CHO-K1 cell line (Chinese hamster ovary), where the administration of doses up to 3000 mg/kg in mice did not reveal signs or symptoms of acute toxicity. Regarding the assessment of chronic toxicity, the groups that were tested with the same dose until the fifth week exhibited weight gain, and the group treated with a 500 mg/kg dose did not exhibit metabolic changes. The treated group exhibited changes in the number of leukocytes. No cytotoxic effect was observed on the CHO-K1 cells since cell viability was not affected in the assay with decreasing concentrations (IC_50_ = >200 µg/mL) of HE. Regarding the antimicrobial effect, except for the Helicobacter pylori (MIC 12.5 µg/mL) and Enterococcus faecalis (MIC 100 µg/mL) strains, the 70% HE of *A. chica* could not inhibit the bacterial and fungal growth of the tested species. Furthermore, a quantitative phytochemical analysis of ACE mainly revealed the presence of the phenolic compounds, among which were flavones and flavonols (4.02%), flavanones and dihydroflavonols (0.11%), and anthocyanins (0.06%), as well as the pigments, lycopene, and β-carotene [49].

The leishmanicidal activity of FH of ACE was tested against *Leishmania amazonensis* and *L. infantum*, and the most active fraction (B2) revealed MIC values of 37.2 and 18.6 μg/mL, respectively, causing a significant reduction of the peptidases in both species. Moreover, compared with the untreated parasites, relevant mitochondrial alterations were observed in the promastigote forms of *L. infantum*. The phytochemical analysis of B2 mainly revealed the presence of the following compounds: linolenic acid, methyl ester, n-hexadecanoic acid, octadecanoic acid, and gammasitosterol [50]. 

Like osteoarthritis, tendinopathy has been treated with traditional medicines, such as *A. chica*. It exerted healing effects on the tendons of seven separate groups of animals, following partial transection. The topical application of the extract during the healing process stimulated the synthesis of total collagen between the 7th and 21st day, thus proving the effectiveness of the raw extract of *A. chica* in wound healing [51]. 

A previous study [52] also demonstrated the efficiency of *A. chica* in the treatment of anemia, inflammation, intestinal colic, hepatitis, and skin infections. The research group evaluated the crude aqueous extract and the fractions of *Arrabidaea* in Salmonella tests to investigate the mutagenic potential of the chemical compounds in *A. chica*. However, the compounds did not exhibit any mutagenic response to Salmonella strains, as well as to genotoxicity in hamster ovary cells, owing to the phytochemical composition of known anti-inflammatory compounds with high concentrations.

As previously reported, the extract of *A. chica* is applied to traditional medicine to induce several biological activities, particularly anti-inflammatory, antimicrobial, diuretic, and antioxidant activities, as well act as a wound-healing agent. The DCM fraction exhibited excellent antioxidant activity, and this result is due to the presence of flavonoids, mainly (**26**). Studies have confirmed that *Arrabidaea chica* can also be employed for the treatment of urinary infection [35]; the ethanolic and hexane extracts obtained positive results regarding diuretic activity with different doses.

The application of nanotechnology to extracts greatly benefits phytotherapy, thus emphasizing the advantages of nanotechnology, such as supply, bioavailability, and pharmaceutical products. The authors [53] reported the preparation and characterization of NPs (polymeric nanoparticles), which together with the extract, induced cell viability at very low concentrations. The antiulcerogenic activity was also evaluated, and the result demonstrated that the fusion of NPs with the extract could reduce the ulcerative lesion, making it possible to encapsulate the extract in NPs.

The Brazilian Amazonian population employs *A. chica* leaves to traditionally treat anemia, although the mineral content of this species is unknown. The authors [54] determined the elemental mineral contents (Cu, Fe, Mn, and Zn) of leaves and teas that were obtained via the decoction and infusion of three varieties of the species (AC1, AC2, and AC3) via flame atomic absorption spectrometry. Generally, all the varieties of the studied species exhibited mineral levels, although AC2 exhibited higher concentrations of all the studied minerals and forms of extraction. Fe was more abundant in the dried leaves (38.4–115.5 µg/g), while Mn was the most abundant in decoction (56.1–62.7%) and infusion (45.6–63.6%) extractions.

Further, *A. chica* leaves are utilized by indigenous groups to clean wounds and treat fungal infections. The cytotoxicity test of CHO-K1 demonstrated that HEFc and (**38a**) did not exhibit cytotoxicity with IC_50_ values of >200 and >40 µg/mL, respectively, while the positive control (doxorubicin) was highly cytotoxic with IC_50_ of 0.30 ± 0.04 µg/mL. HEFc and the fraction containing (**33b**) exhibited good activity against the tested bacteria, while (**38a**) did not exhibit activity. The mechanism of the antibacterial action of HEFc was studied with the nucleotide leakage assay in a medium containing *S. pyogenes* (MIC, 12.5 µg/mL). HEFc caused leakage in the order of 88.9% within 12 h and 90.0% leakage when associated with amoxicillin [40].

The ethanolic crude extract, as well as chloroform (FC), methanol, and EAA fractions of *A. chica* leaves and stems, were evaluated for their cytotoxicity against *Leishmania amazonensis* promastigotes at 24, 48, and 72 h intervals, followed by their healing effect on the surgical skin lesions of Swiss mice. The extract exhibited a 50% reduction in the viability of *L. amazonensis* promastigotes at a concentration of 125 μg/mL and the tested incubation intervals. The FC, methanol, and EAA fractions also exhibited a 50% reduction in viability at concentrations of 120, 120, and 60 μg/mL, respectively. EAA acted against the parasite at a lower concentration than the extract. Furthermore, the phytochemical screening of the crude extract revealed the presence of flavonoids, tannins, anthocyanidins, and chalcones. Regarding the healing activity, the extract exhibited satisfactory responses during angiogenesis and collagen deposition; however, the extract did not accelerate the healing of the lesions possibly because of local and systemic factors [55].

The quantitative analysis indicated that 8.77 ± 0.23 mg/g of the compound (**39**) was present in the extract. The oral pretreatment of LPS-induced peritonitis in mice employing HEFc and (**39**) achieved the decreased migration of leukocytes into the peritoneal cavity by 44.0% at a single dose of 20 mg/kg of the extract, as well as 56.0% and 67.0% for doses of 4 and 20 mg/kg of (**39**), respectively, in addition to the reduced concentrations of pro-inflammatory cytokines (TNFa and IL-1b) [41].

The carotenoid and phenolic compound profiles of the freeze-dried leaves and hydromethanolic extract of *A. chica* were determined via HPLC-DAD-MS, and the main compounds identified include (**38a**) (15,147.22 μg/g, dry basis, b.d.), (**27**) (204.28 μg/g, b.d.), (**40a**) (129.5 μg/g, b.d.), and (**40b**) (79.86 μg/g, b.d.). Additionally, the presence of ascorbic acid (152.7 μg/g) in *A. chica* leaves was reported for the first time. The extract containing 86.81 μM Trolox/g of the fresh leaves was also demonstrated as an excellent ABTS free-radical scavenger; it inhibited the degradation of tryptophan by ^1^O_2_ at all concentrations and exerted a concentration-dependent effect at an IC_50_ value of 177 μg/mL [36].

The potential mutagenic and genotoxic/antigenotoxic effects of CF of *A. chica* leaves (Ac–CF) was investigated and showed that they did not exert any mutagenic effect on the tested Salmonella typhimurium strains at any of the concentrations (5 and 250 µg Ac–CF/plate). Furthermore, Ac–CF exhibited negative responses regarding mutagenicity and genotoxicity in mice and exerted an antigenotoxic effect by decreasing hydrogen peroxide-induced oxidative DNA damage by >50% in their blood and livers [56].

ACE was still employed to evaluate its antitumor effect that is associated with a chemotherapeutic agent, which is employed in cancer treatment (vincristine), as well as its ability to attenuate the development of DMBA-induced breast carcinoma in rats. Fluorescence imaging confirmed that ACE alone or ACE combined with vincristine (50%) reduced the presence and volume of breast tumors compared with the control group. The vincristine-treated group exhibited higher levels of ALT, AST, and GGT (parameters that are associated with high vincristine toxicity). However, a significant reduction was observed in the liver and hematologic toxicity of the ACE + 50% vincristine-treated group. Furthermore, the ACE-treated rats exhibited better albumin and transaminase levels probably because of the antioxidant components of ACE [57].

The leishmanicidal activities of anthocyanidins from four morphotypes of *A. chica* that were collected during summer and winter were evaluated. The multivariate analysis of the morphotypes revealed the presence of (**33a**), (**33b**), (**33c**) and (**33d**). Further, two different groups were recognized: Group I containing (33c) and Group II with a high (**33a**) content. The morphotype ACIV(S) extract, which is rich in carajurin, exhibited the best leishmanicidal activity among all the extracts that were analyzed (IC_50_ = 101.5 µg/mL) [42].

The physicochemical and biological characteristics of a flexible film containing chitosan and alginate that were incorporated into standardized ACE, Verlot, were evaluated to verify the influence of adding a surfactant and silicone-based compounds to obtain a dense and porous flexible structure as a device for releasing bioactive compounds into lesions. The incorporation of a surfactant and silicone gel ensured that a porous matrix was obtained, which also increased the intumescence of the polymeric matrix. The formulations containing a surfactant exhibited better performances regarding flexibility and deformation to rupture. Additionally, the presence of the standardized ACE did not negatively affect the characteristics of the membranes; it demonstrated release concentration values of 89–99 μg/mL, which are within the therapeutic range [58].

The antioxidant capacities of the FH, FC, and EAA from the leaf extract (EC) of *A. chica* were evaluated with 2,2-diphenyl-1-picrylhydrazyl (DPPH) and the xanthine/luminol/xanthine oxidase system. FC exhibited the best antioxidant activity among the other fractions (EC_50_ = 7.3 and 0.33 μg/mL). Moreover, FC and CE exerted photoprotective effects by reducing ultraviolet (UV)-induced cell damage, inhibiting ROS production, and preventing lipid peroxidation [59].

Next, the photoprotective activity and toxicity level of the formulations based on a non-ionic cream containing the extract (5.0%), as well as FH and EAA (2.50%) obtained from *A. chica* leaves, were evaluated. The extract and fractions exhibited good optical absorption in the UV regions, including UVA and UVB. The formulation obtained without adding the inorganic UV filters that are commonly employed in commercial sunscreens exhibited an ex vivo percutaneous penetration rate similar to that of the positive control, as well as the ability to remain on the skin for 180 min. The macroscopic and histopathological analysis did not reveal any signs of irritation and significant alterations to the skins of the groups that were treated with the formulation; moreover, no renal or hepatic toxicities were observed similar to the control group [60].

The intestinal anti-inflammatory activity of HEFc was investigated. The group, which was treated with the plant extract (5-fluorouracil), exhibited partially preserved villi and crypts, reduced infiltration, and preserved muscle layers; the experimental intestinal mucositis model exhibited reduced weight changes [34].

The hydroalcoholic ACE exhibited cytoprotection over the epithelial and osteoblastic cells that were exposed to zoledronic acid. The cells which were treated with the extract exhibited increased cell viability with values of 74.10–82.30% at concentrations of 5 and 10 μg/mL for fibroblasts and 66% for the pre-osteoblasts and a lower caspase 3/7 [61].

#### 3.1.7. *A. patellifera*

Synonyms: *Arrabidaea patellifera* (Schltdl.) Sandwith, *Fridericia patellifera* (Schltdl.) L.G. Lohmann, *Anemopaegma cupulatum* (Splitg.) Bureau & K. Schum., *Arrabidaea broadwayi* (Sprague & Riley) Sandwith, *Arrabidaea patellifera* (Schltdl.) Sandwith, *Bignonia patellifera* (Schltdl.), *Cuspidaria mollis* Kränzlin, *Micropaegma cupulatum* (Splitg.) Pichon, *Parmentiera patellifera* (Schltdl.) Miers, *Petastoma breviflorum* Standl., *Tabebuia neurophylla* Miq., *Tanaecium paniculatum* Siebert ex DC.

##### Main Chemical Composition of *A. patellifera*

The chemical investigation of the methanolic extract of *A. patellifera* leaves, a Bignoniaceae from Panama, obtained the isolation of mangiferin (**42a**), and six mangiferin derivatives, namely 3′*-O*-*p*-hydroxy benzoyl mangiferin (**42b**), 3′-*O*-trans-coumaroyl-mangiferin (**42c**), 6′-*O*-trans–coumaroyl-mangiferin (**42d**), 3′-*O*-trans-cinnamoylmangiferin (**42e**), 3′-*O*-trans-caffeoylmangiferin (**42f**), and 3’-*O*-benzoylmangiferin (**42g**) (Table 7) [62].

##### Bio-Pharmacological Properties of *A. patellifera*

The compounds (**42a**–**42g**) exhibit antioxidant and radical scavenging activities, and four of them were relatively active in vitro against *Plasmodium falciparum* [62].

The reports of chemical, biological and pharmacological aspects from seven species of Arrabidaea are sufficient to valorize this genus. These species revealed a great diversity of unknown and bioactive compounds with varied molecular patterns, with xanthones the typical type of compound to the genus, since they were found in different species. 

The healing properties reported may also be associated with activities of isolated compounds reported in Arrabidaea genus. The literature reports the isolation of several secondary metabolites in species from this genus distributed in tropical and subtropical regions all over the world. These studies showed the occurrence of substances of various other classes with triterpenes, flavonoids, and other phenolic compounds. Studies of ethnopharmacological data validation confirmed several popular indications to a lot of species of Arrabidaea, including antibacterial, antifungal, anti-inflammatory, and antioxidant activities. 

The continuous identification of new metabolites from Arrabidaea, allied to new methods of pharmacological and biological evaluation in projects of bioprospection of drugs, is determinant in the revaluation of several vegetal extracts, aiming at the search for new chemical classes and the identification of active metabolites, being rather selective to certain biological targets.

Obviously, members from the Arrabidaea genus have significant potential with promising pharmacological activities of extracts in the context of traditional medicine, especially in the field of tropical diseases, skin issues, and venereal diseases. Pharmacological studies must be performed considering the traditional method of preparation and application. 

## 4. Conclusions

The genus Arrabidaea contains abundant phytoconstituents, mainly comprising flavonoids, triterpenes, and xanthones. Moreover, the genus Arrabidaea exhibits significant pharmacological potentials and promising activities, such as antioxidant, antimicrobial, anticancer, antileishmanial, antiprotozoal, and anti-inflammatory activities. The knowledge obtained from this review can increase the efficacy of the already employed traditional applications of these plants and promote their cultural values afterward. This review also stimulates chemical and bio-pharmacological studies with the genus Arrabidaea and serves as an alert for the elaboration of strategies for the preservation of Brazilian biodiversity.

## Figures and Tables

**Figure 1 pharmaceuticals-15-00658-f001:**
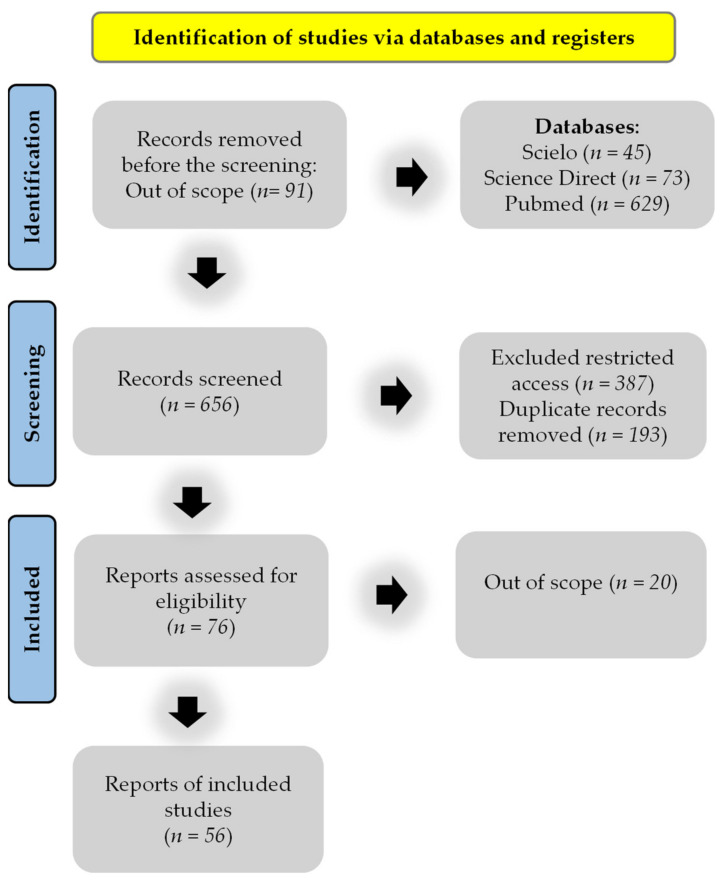
Flow diagram of the reference screening and inclusion.

**Table 1 pharmaceuticals-15-00658-t001:** Structure and bio-pharmacological properties of isolated compounds from *A. samydoides*.

N°	Compound Structure and Name	Species Name	Plant Part, Solvent	Biological Effect	Ref.
**1**	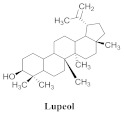	*A. samydoides*	Leaves and stem, EtOH	Not evaluated	[1]
**2**	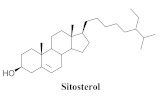	*A. samydoides*	Leaves and stem, EtOH	Not evaluated	[1]
**3**	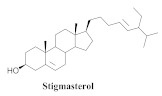	*A. samydoides* and *A. bilabiata*	Leaves and stem, EtOH	Not evaluated	[1]
**4**	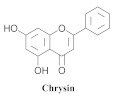	*A. samydoides*	Leaves and stem, EtOH	Not evaluated	[1]
**5a**	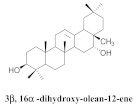	*A. samydoides*	Leaves and stem, EtOH	Not evaluated	[1]
**5b**	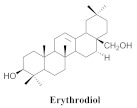	*A. samydoides*	Leaves and stem, EtOH	Not evaluated	[1]
**6a**	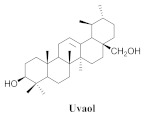	*A. samydoides* and *A. pulchra* and *A. triplinervia* and *A. bilabiata*	Leaves and stem, EtOH	Not evaluated	[1]
**6b**	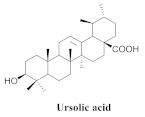	*A. samydoides*	Leaves and stem, EtOH	Trypanocidal	[1,14,15,16]
**7a**	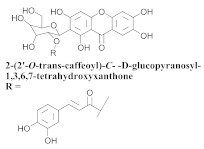	*A. samydoides*	Stem, EtOH	Antioxidant	[10]
**7b**	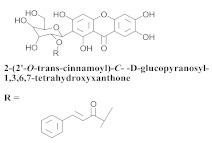	*A. samydoides*	Stem, EtOH	Non active	[10]
**7c**	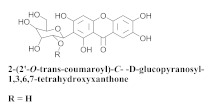	*A. samydoides*	Stem, EtOH	Non active	[10]
**7d**	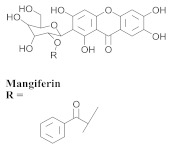	*A. samydoides*	Stem, EtOH	Non active	[10]
**7e**	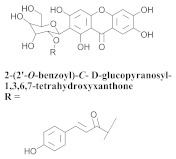	*A. samydoides*	Stem, EtOH	Non active	[10]
**7f**	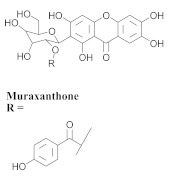	*A. samydoides*	Stem, EtOH	Antioxidant	[10]

**Table 2 pharmaceuticals-15-00658-t002:** Structure and bio-pharmacological properties of isolated compounds from *A. pulchra*.

N°	Compound Structure and Name	Species Name	Plant Part, Solvent	Biological Effect	Ref.
**8a**	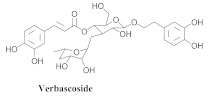	*A. pulchra*	Leaves, EtOH	Antiviral	[14]
**8b**	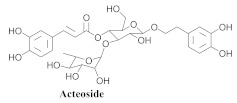	*A. pulchra*	Aerial parts, EtOH	15-LOX inhibitory	[18]
**9**	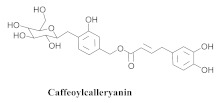	*A. pulchra*	Leaves and aerial parts EtOH	Antiviral and 15-LOX inhibitory	[14,18]
**10**	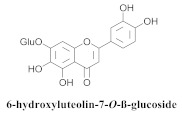	*A. pulchra*	Aerial parts, EtOH	15-LOX inhibitory	[18]
**11**	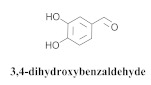	*A. pulchra*	Aerial parts, EtOH	Non active	[15]
**12**	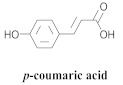	*A. pulchra*	Aerial parts, EtOH	Non active	[15]
**13**	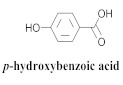	*A. pulchra*	Aerial parts, EtOH	Non active	[15]
**14**	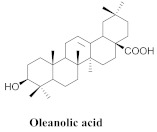	*A. pulchra* and *A. triplinervia*	Aerial parts, EtOH	Trypanocidal	[15,16]

**Table 3 pharmaceuticals-15-00658-t003:** Structure and bio-pharmacological properties of isolated compounds from *A. triplinervia*.

N°	Compound Structure and Name	Species Name	Plant Part, Solvent	Biological Effect	Ref.
**15**	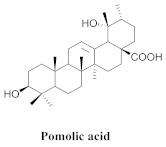	*A. triplinervia*	Leaves, EtOH	Non active	[16]
**16**	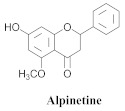	*A. triplinervia*	Leaves, EtOH	Non active	[16]

**Table 4 pharmaceuticals-15-00658-t004:** Structure and bio-pharmacological properties of isolated compounds from *A. bilabiata*.

N°	Compound Structure and Name	Species Name	Plant Part, Solvent	Biological Effect	Ref.
**17**	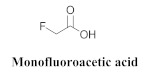	*A. bilabiata*	Seeds and leaves, aqueous extract	Not evaluated	[21]
**18**	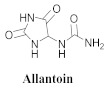	*A. bilabiata* and *A. chica*	Aerial parts, IsopOH	Motor activity	[20,22]

**Table 5 pharmaceuticals-15-00658-t005:** Structure and bio-pharmacological properties of isolated compounds from *A. brachypoda*.

N°	Compound Structure and Name	Species Name	Plant Part, Solvent	Biological Effect	Ref.
**19a**	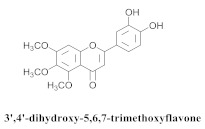	*A. brachypoda*	Leaves, chloroform	Fungicide	[7]
**19b**	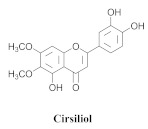	*A. brachypoda*	Leaves, chloroform	Fungicide	[7]
**19c**	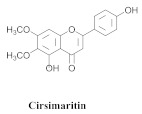	*A. brachypoda*	Leaves, chloroform	Fungicide	[7]
**19d**	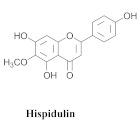	*A. brachypoda* and *A. chica*	Leaves, chloroform	Non active	[7,34]
**20**	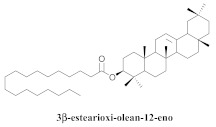	*A. brachypoda*	Roots, EtOH	Anti-inflammatory	[26]
**21a**	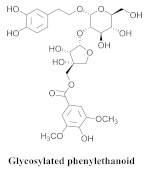	*A. brachypoda*	Roots, EtOH	Not evaluated	[11]
**21b**	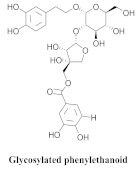	*A. brachypoda*	Roots, EtOH	Not evaluated	[11]
**22a**	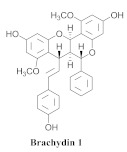	*A. brachypoda*	Roots, EtOH	Antiproliferative	[11,27,28,29]
**22b**	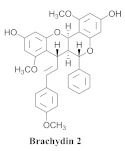	*A. brachypoda*	Roots, EtOH	Leishmania, Antiproliferative and arthritis	[11,27,28,29]
**22c**	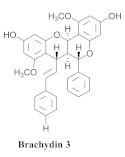	*A. brachypoda*	Roots, EtOH	Antiproliferative and arthritis	[11,27,28]
**22d–j**	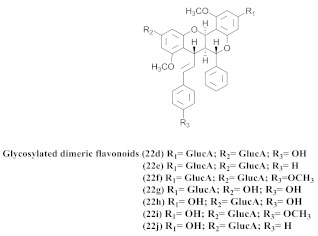	*A. brachypoda*	Roots, EtOH	Gastroprotective	[11]
**23a–p**	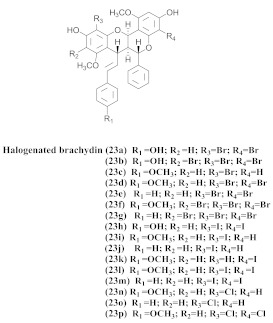	*A. brachypoda*	Roots, EtOH	Trypanocidal (23e, 23k, 23n, 23o) and Leishmanicidal (23g and 230)	[12]
**24**	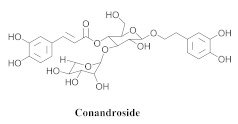	*A. brachypoda*	Aerial parts, EtOH	Inhibition of LOX	[30]
**25**	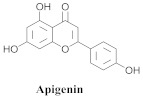	*A. brachypoda* and *A. chica*	Leaves, EtOH	Not evaluated	[31,35]
**26**	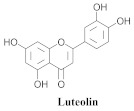	*A. brachypoda* and *A. chica*	Leaves and roots, EtOH	Antiproliferative and diuretic	[31,33,35,36]
**27**	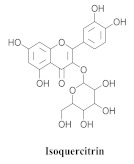	*A. brachypoda* and *A. chica*	Leaves, EtOH	Not evaluated	[31]
**28**	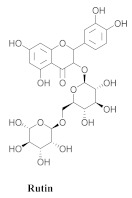	*A. brachypoda*	Leaves, EtOH	Not evaluated	[31]
**29**	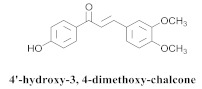	*A. brachypoda*	Flowers, EtOH	Non active	[32]
**30**	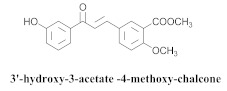	*A. brachypoda*	Flowers, EtOH	Non active	[32]
**31**	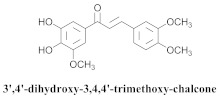	*A. brachypoda*	Flowers, EtOH	Non active	[32]
**32**	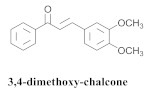	*A. brachypoda*	Flowers, EtOH	Antimicrobial	[32]

**Table 6 pharmaceuticals-15-00658-t006:** Structure and bio-pharmacological properties of isolated compounds from *A. chica*.

N°	Compound Structure and Name	Species Name	Plant Part, Solvent	Biological Effect	Ref.
**33a**	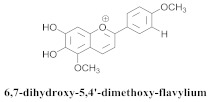	*A. chica*	Leaves, Et_2_O	Inhibited NF-kB	[8,42]
**33b**	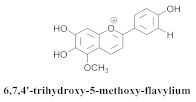	*A. chica*	Leaves, Et_2_O	Non active	[8,40]
**33c**	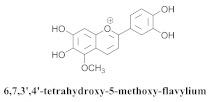	*A. chica*	Leaves, Et_2_O	Non active	[8,42]
**33d**	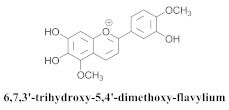	*A. chica*	Leaves, Et_2_O	Non active	[8,42]
**34**	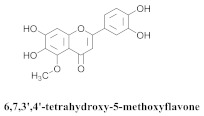	*A. chica*	Leaves, MeOH	Non active	[39]
**35**	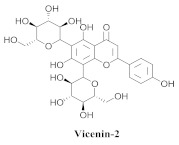	*A. chica*	Leaves, EtOH	Not evaluated	[38]
**36**	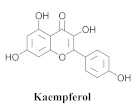	*A. chica*	Leaves, EtOH	Not evaluated	[38]
**37**	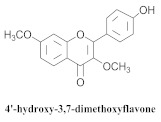	*A. chica*	Leaves, EtOH	Not evaluated	[38]
**38a**	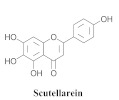	*A. chica*	Leaves, EtOH	Non active	[34,36,40]
**38b**	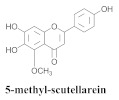	*A. chica*	Leaves, EtOH	Not evaluated	[34]
**39**	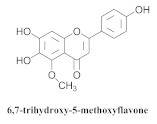	*A. chica*	Leaves, EtOH	Anti-inflammatory	[41]
**40a**	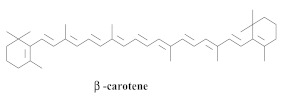	*A. chica*	Leaves, Acetone	Not evaluated	[36]
**40b**	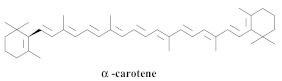	*A. chica*	Leaves, Acetone	Not evaluated	[36]
**41a**	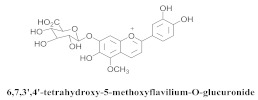	*A. chica*	Leaves, EtOH	Not evaluated	[34]
**41b**	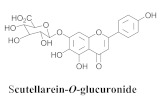	*A. chica*	Leaves, EtOH	Not evaluated	[34]
**41c**	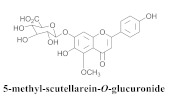	*A. chica*	Leaves, EtOH	Not evaluated	[34]

**Table 7 pharmaceuticals-15-00658-t007:** Structure and bio-pharmacological properties of isolated compounds from *A. patellifera*.

N°	Compound Structure and Name	Species Name	Plant Part, Solvent	Biological Effect	Ref.
**42a**	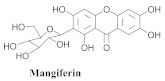	*A. patellifera*	Leaves, MeOH	Antioxidant and *Plasmodium falciparum*	[62]
**42b**	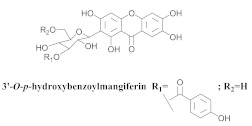	*A. patellifera*	Leaves, MeOH	Antioxidant and *Plasmodium falciparum*	[62]
**42c**	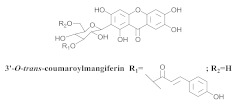	*A. patellifera*	Leaves, MeOH	Antioxidant and *Plasmodium falciparum*	[62]
**42d**	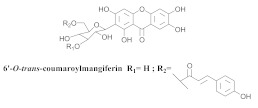	*A. patellifera*	Leaves, MeOH	Antioxidant and *Plasmodium falciparum*	[62]
**42e**	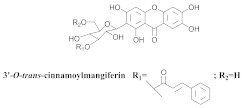	*A. patellifera*	Leaves, MeOH	Antioxidant	[62]
**42f**	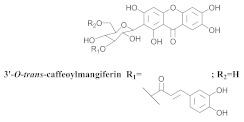	*A. patellifera*	Leaves, MeOH	Antioxidant	[62]
**42g**	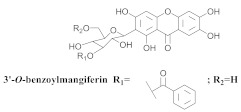	*A. patellifera*	Leaves, MeOH	Antioxidant	[62]

## Data Availability

Data sharing not applicable.

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
