# Peer review of "A Review of the Phytochemistry and Pharmacological Properties of the Genus Arrabidaea"

_pharmaceuticals, 2022, doi:10.3390/ph15060658_

Round 1
Reviewer 1 Report
In this paper, the botanical characteristics, chemical composition, and pharmacological properties of Arrabidaeae species have been evaluated. The description of chemical, biological, and pharmacological features from seven Arrabidaea species is adequate to bring this genus in the focus. The overall idea of this article is clear, however, some major issues need carefully revision for publication. The specific comments are as follows for the authors' reference.
- A little background should be added at the beginning of the abstract.
- Some content in the abstract is repeated, so some words can be removed, and the order of sentences adjusted to make sense.
- In line 37, "," that is between “Pharmacological activity” and “Arrabidaea” should be changed to ";".
- It is suggested that some unnecessary descriptions, such as the distribution and horticultural applications of Bignoniaceae species, be removed from the "Introduction." Furthermore, why can the "Introduction" be detailed separately from -lapachone?
- The structure of the main text should be more refined.
- In section 2.2, the contents of lines 128-130 and 131-132 are repeated.
- In line 147, “bio/pharmacological” should be changed to “bio-pharmacological”.
- The "Materials and Methods" section should be moved below the "Introduction."
- In the discussion part, it's a good option to include some opinions and thoughts.
- The order between the paragraphs in the "Introduction" and "Results and discussion" is confusing, and it is recommended to revise it to fit the overall logic of the article. The highlight of the article should be prioritized.
- It is suggested that certain implications and innovations from this study be included to the "Results and Discussion" section.
- It is suggested not to barely summarize in the end of "Results and discussion" but to add tables of bio-pharmacological properties and isolated compounds with their structures separately after the review of each species to make it clearer.
- It is suggested to show the pharmacological data of each species in the form of chart to reflect their pharmacological action.
- It is suggested to summarize the extraction method of the substances isolated from the genus Arrabidaea.
Reviewer 2 Report
Nascimento et al. reported a "Review of the chemical and pharmacological aspects of Arrabidaea species". They sorted out the chemical compositions and their health benefits. The article severely lacks its purpose, and the methodology does not support the report, as it is incomplete. It seems this article is a systematic review, but there is no details methodology mentioned and no PRISMA flow chart included. The authors are advised to go through the http://www.prisma-statement.org/ and report this review according to their guidelines as well as include the PRISMA Checklist as a supplementary file. Other specific comments are below:
The genus name in the title should be italicized. Aspect is not a suitable word for the title. The title can be rewritten as "A systematic review of the phytochemistry and pharmacological properties of the genus Arrabidaea".
Overall the abstract is not well written; rather, it is just a random presentation of some information about the Arrabidaea. The abstract need to be rewritten. Answering the following question might help rewrite the abstract: What was done? Why did you do it? What did you find? Why are these findings useful and important?
Line 19-21: Are you sure the mentioned health benefits belong to all species of Arrabidaea? If not, please specify the species name.
Line 40-41: The number of species is not mentioned in the cited article (Ref. 1). Cite this information from the original source. Additionally, ref. 1 is not the original source of this sentence. The author should cross-check the original source and cite the original source.
Line 43-44: Cite the information.
Line 44-45: Bignoniaceae is not a species. It's a family and italicized throughout the manuscript where it appeared.
Line 53-54: Bignoniaceae is not a species. Correct it.
Line 56-61: Be specific while writing the chemical constituents. Because it diversely varied from species to species. Also, write the genus and species name correctly.
Line 64-65: cite the source
Line 73-83: The authors mentioned the previous studies on Arrabidaea spp. So, how does the current study differ from the existing studies? The authors need to clarify this in the introduction.
The introduction lacks problem statements and how they are going to solve this in this manuscript.
Section 2.1: It deems not suitable in this section. The authors can concisely mention this in the introduction.
2.2 Botanical characterization of the Arrabidaea genus. Either write Arrabidaea spp. Or genus Arrabidaea.
In this section, the authors can cite an appropriate source that mentions all the species of Arrabidaea, or they can supply it as a supplementary file. Because knowing all species of Arrabidaea is important, and why some of the species are important for further study would be justifiable.
Section 2.3 The authors only selected 7 species from the studied genus Arrabidaea but why only 7? The authors mentioned genus Arrabidaea has ~170 species. Among these, only 7 do not represent the whole genus. It's a selective study. So, the authors need to discuss it and the reason for choosing 7 species only. In such a case, the authors need to change the title too. Additionally, the authors can present data, including the following subheading for comprehensive presentations.”
Plant species:
Synonyms:
Brief plant description:
Lead phytocompounds:
Pharmacological properties: initially, authors can highlight the available pharmacological activities and briefly discuss the salient findings from in vitro, in vivo and clinical studies.
The authors can follow this for all the selected species to highlight the findings.
Table 1. It needs to be more comprehensive. The way presented information is vague. Because sure all the mentioned compounds do not show all the activities. So, be specific about each activity. You can use the separate row for each activity or name one compound and list all of its possessing activities. The authors can mention the compound isolation part of the plant, dose, experimental model and key results. Also, is there any cytotoxicity? Highlighting this is very crucial for pharmacological study.
In the Phytochemistry section, arrange them based on the species-specific. For example, compounds A, B and C are found in species X,Y, Z, but compound D is only found in Y. Arranging them appropriately will get rid of the biases of data presentation.
Methodology rewrites as per general comments initially and follows the PRISMA guidelines. And appropriately write the search keywords and search database names. Clearly mention the inclusion and exclusion criteria. In the abstract, you mentioned pure substances isolated since 1927, but you included studies only between 1994-and 2021. Please correct it.
Round 2
Reviewer 1 Report
The revised article is overall more logical and more clearly structured. However, there are still some issues. The following suggestions are for the authors' reference.
- In the introduction, it is recommended that the description of ecological and economic considerations about the genus Arrabidaea be reduced appropriately. The introduction section should be concise and straightforward.
- It might be helpful to end the introduction with the reasons for selecting 7 species from the genus Arrabidaea of the study.
- Section 3.1 might be better placed in the introduction after streamlining it, which is a little more suitable in section 3 because of the extensive exposition of section 3.2.
- In section 3.2, it is suggested that adding some subheadings to make the structure clearer. For example, 3.2.1 Arrabidaea samydoides, 3.2.1.1 Main chemical composition of Arrabidaea samydoides, 3.2.1.2 Bio-pharmacological properties of Arrabidaea samydoides.
Reviewer 2 Report
The authors tried to answer the comments, but the manuscript still lacks significantly, particularly methods and discussion. Results also need to be categorized as "Studies characterizations". I don't fill this review contributed significant knowledge to the scientific community. Therefore, my recommendation is against publication with the current version.
